# Mechanical, Histological, and Scanning Electron Microscopy Study of the Effect of Mixed-Acid and Heat Treatment on Additive-Manufactured Titanium Plates on Bonding to the Bone Surface

**DOI:** 10.3390/ma13225104

**Published:** 2020-11-12

**Authors:** Naoko Imagawa, Kazuya Inoue, Keisuke Matsumoto, Ayako Ochi, Michi Omori, Kayoko Yamamoto, Yoichiro Nakajima, Nahoko Kato-Kogoe, Hiroyuki Nakano, Tomiharu Matsushita, Seiji Yamaguchi, Phuc Thi Minh Le, Shinpei Maruyama, Takaaki Ueno

**Affiliations:** 1Division of Medicine for Function and Morphology of Sensor Organs, Department of Dentistry and Oral Surgery, Faculty of Medicine, Osaka Medical College, 2-7 Daigakumachi, Takatsuki, Osaka 569-8686, Japan; ora095@osaka-med.ac.jp (N.I.); ora077@osaka-med.ac.jp (K.M.); ora083@osaka-med.ac.jp (A.O.); ora085@osaka-med.ac.jp (M.O.); ora071@osaka-med.ac.jp (K.Y.); n4160@osaka-med.ac.jp (Y.N.); ora078@osaka-med.ac.jp (N.K.-K.); ora099@osaka-med.ac.jp (H.N.); ueno@osaka-med.ac.jp (T.U.); 2Department of Biomedical Science, College of Life and Health Sciences, Chubu University, 1200 Matsumoto-cho, Kasugai, Aichi 487-8501, Japan; matsushi@isc.chubu.ac.jp (T.M.); sy-esi@isc.chubu.ac.jp (S.Y.); minhphuc@isc.chubu.ac.jp (P.T.M.L.); 3Osaka Yakin Kogyo Co., Ltd., 4-4-28, Zuiko, Yodogawa-ku, Osaka 533-0005, Japan; maruyama@e.osakayakin.co.jp

**Keywords:** titanium, bone bonding, selective laser melting, mixed-acid and heat treatment, additive manufacturing, custom-made plate

## Abstract

The additive manufacturing (AM) technique has attracted attention as one of the fully customizable medical material technologies. In addition, the development of new surface treatments has been investigated to improve the osteogenic ability of the AM titanium (Ti) plate. The purpose of this study was to evaluate the osteogenic activity of the AM Ti with mixed-acid and heat (MAH) treatment. Fully customized AM Ti plates were created with a curvature suitable for rat calvarial bone, and they were examined in a group implanted with the MAH-treated Ti in comparison with the untreated (UN) group. The AM Ti plates were fixed to the surface of rat calvarial bone, followed by extraction of the calvarial bone 1, 4, 8, and 12 weeks after implantation. The bonding between the bone and Ti was evaluated mechanically. In addition, AM Ti plates removed from the bone were examined histologically by electron microscopy and Villanueva–Goldner stain. The mechanical evaluation showed significantly stronger bone-bonding in the MAH group than in the UN group. In addition, active bone formation was seen histologically in the MAH group. Therefore, these findings indicate that MAH resulted in rapid and strong bonding between cortical bone and Ti.

## 1. Introduction

Bone defects caused by trauma and tumor in the maxillofacial region require highly advanced reconstructive surgery, such as free vascularized autologous bone grafts using vascular anastomoses of the tibia and scapula. In clinical practice, autologous grafting of bones, including the tibia, fibula, and femur, has been performed for the reconstruction of hard tissues, such as bones lost due to trauma, inflammation, and tumor, and good outcomes have been reported [1,2,3]. However, this requires highly advanced surgical techniques, the amount of autologous bone used is limited, and the donor site surgical infection is concerned. Therefore, artificial bone with shape-recovery capacity and high compatibility with the living body has long been needed to solve these problems, and customized artificial bone, in which hard tissues are artificially created based on computed tomography (CT) scans of a patient, has been in development [4,5,6]. In particular, titanium (Ti), which has high biocompatibility, has been used in many implantable medical devices in clinical settings, including artificial joints and dental implants, and its high efficacy and safety have been demonstrated [7,8,9,10,11,12]. Of the customizable technologies, additive manufacturing (AM) using the selective laser technique, which creates artificial bone with Ti particles based on CT scans of a patient, has attracted attention [13,14,15]. For example, Inoue et al. [16] reported that the use of AM Ti plates was effective for precise bone reconstruction of maxillary bone defects, and Bartolomeu et al. showed functional recovery using similar AM Ti artificial joints [17,18]. Therefore, the AM Ti artificial bone is considered a highly effective device in clinical practice.

Furthermore, a report by Kokubo et al. [19] suggested that Ti exhibits strong bone-bonding when its surface undergoes mixed-acid and heat (MAH) treatment. Ti devices, which are conventionally screws, have played a role in functional recovery by mechanically bonding to the bone. However, if the AM Ti plates can be bonded to the bone, it can minimize the risk of device fracture and infection caused by loosening of the fixation part. Yet such reconstructive plates have not been fully investigated, and a past study [20] evaluated the changes in the bonding between the bone and Ti plates treated with MAH by inserting the Ti plates in rabbit femora. However, changes between the Ti and bone surface were not clarified. Thus, mechanical and histological evaluations of the MAH-treated AM Ti plate and the bone surface are needed. We considered that the AM Ti plate with MAH treatment perform strong bone bonding and early bone formation, and that MAH treatment is an effective approach for prevention of the reconstructive plate fracture and surgical site infection.

In this study, an AM Ti plate treated with MAH was fixed to the surface of rat calvarial bone, and the changes in its bone bonding over time were examined mechanically. In addition, the bone bonding of the AM Ti plates treated with MAH was evaluated histologically by electron microscopy to clarify part of the process of bone formation.

## 2. Materials and Methods

### 2.1. Experimental Method

Eighty Sprague–Dawley (SD) rats (8-week-old females) were divided into the MAH-treated group (experimental group) and the untreated group (control group). The periods of AM Ti plate implantation were 1, 4, 8, and 12 weeks for both groups (10 rats per period per group). Of the 10 rats, 8 were used for the evaluation of bone bonding, and the other 2 were used for histological observation. This study was approved by the Animal Experiment Committee of Osaka Medical University (Takatsuki, Osaka, Japan, No. 2019-098).

### 2.2. Design of the AM Ti Plate

The calvarial bone of 8-week-old SD rats was measured using micro CT (Latheta LCT200, HITACHI, Tokyo, Japan). The average rat calvarial bone was 150 mm in diameter. Thus, AM Ti plates of 150 mm in diameter suitable for the rat calvarial bone were created using the additive manufacturing technique. The selective laser melting (SLM) technique was used on pure Ti powders with an average particle size of 23.3 µm. The Standard Triangulated Language (STL) data of the AM Ti plate were transferred to the additive manufacturing machine (EOS M270, EOS GmbH Electro Optical Systems, Krailling, Germany), and Ti powders were scanned selectively by the laser beam for their solidification. Additive manufacturing, in which the operation was repeated every 0.3 mm, created the designed samples three-dimensionally [21,22]. As shown in Figure 1, the dimensions of the AM samples were 10 mm × 10 mm × 1 mm, and semicircular hooks for traction were formed at the four corners and the center of the samples.

### 2.3. Mixed-Acid and Heat Treatment

The AM Ti plates were immersed in 5 mL of the mixed-acid solution of a 1:1 (*w/w*) mixture of 66.3% H_2_SO_4_ (*w/w*) solution (FUJIFILM Wako Pure Chemical Corporation Co., Ltd., Osaka, Japan) and 10.6% HCl (*w/w*) solution (FUJIFILM Wako Pure Chemical Corporation Co., Ltd., Osaka, Japan) and swung at 70 °C for 1 h at a rate of 120 strokes/min in a shaking water bath (Personal H(r), Taitec, Koshigaya, Japan). They were rinsed under running ultrapure water for 30 s and dried at 40 °C overnight. Then, the samples were heated to 600 °C in an electric furnace (FO300(r), Yamato Scientific Co., Ltd., Tokyo, Japan) with a programmed rate of 5 °C/min in air. After being maintained at the same temperature for 1 h, they were allowed to cool naturally. After the MAH treatment, the AM Ti plates were sterilized by γ rays.

### 2.4. Implantation Technique

After rats were anesthetized with isoflurane, infiltration anesthesia with 2% xylocaine was administered to their head. An incision was made on their head skin with a No15 surgical blade, and the periosteum was elevated to expose the surface of the calvarial bone. As shown in Figure 2, the AM Ti plate was fixed to the calvarial bone with two microscrews (Φ 1.0 mm × 4 mm KLS Martin, Tuttlingen, Germany). The periosteum (polyglactin suture) (Vicryl®, Johnson & Johnson, New Brunswick, NJ, USA) and skin (black silk thread) were sutured to close the wound.

### 2.5. Evaluation of Bone Bonding

As shown in Figure 3, a device that fixes the rat calvarial bone was designed to maintain the AM Ti plates attached to the bone parallel to the floor during the evaluation of bone bonding, allowing the pressure to be applied evenly to the AM Ti plates. A device with a direction-correction function was developed by adding a bearing mechanism to maintain the AM Ti plates parallel to the floor. By aligning the axis centers of the AM Ti plate and the autograph and applying a tensile force using four wires of the same length, the rotation of the sample table in which tension in the direction perpendicular to the AM Ti plate acts due to rotation in the X-axis direction and rotation in the Y-axis direction makes the AM Ti plate parallel to the floor. The calvarial bone was extracted from each rat, which had been euthanized with an anesthetic overdose, together with the AM Ti plate 1, 4, 8, and 12 weeks after its implantation. The AM Ti plate was fixed in super hard gypsum within 24 h of bone extraction and connected to the bone-bonding evaluation device with a bearing function that maintained the AM Ti plate parallel to the floor. A 0.3-mm metal ligature was passed through the semicircular hooks of the sample and connected to the autograph (Shimadzu, AG-I 100kN, Kyoto, Japan) via the evaluation device, and the ascending speed of the autograph was set to 1.0 mm/min. As shown in Figure 3, the bonding between the AM Ti plate and the cortical bone was evaluated by measuring the fracture load (N) as the AM Ti plate was pulled out from the calvarial bone.

### 2.6. Scanning Electron Microscope Observation

The AM Ti plate removed from the calvarial bone was fixed with 20% formalin, dehydrated stepwise with ethanol at different concentrations (70%, 80%, 90%, and 99.5%), and then dried in a vacuum (24 h). The AM Ti plate surface that had been in contact with the calvarial bone was carbon-coated, and scanning electron microscopic observation and elemental analysis by energy dispersive X-ray spectrometer were performed.

### 2.7. Histological Observation

The tissue samples were fixed with a 20% formalin solution and subsequently dehydrated/defatted with ethanol and acetone. Thereafter, the tissue samples were immersed in methyl methacrylate (MMA) to prepare MMA resin blocks. Frontal sections were prepared by cutting the block using a micro-cutting machine (BS-300C/EXAKT, Meiwafosis Co., Ltd. Tokyo, Japan) and a micro-grinding machine (MG-400CS/EXAKT Meiwafosis Co., Ltd. Tokyo, Japan) to pass through the center of the bone defect created. Ground slides (approx. 40 mm thick) were prepared. Finally, the 40-μm-thick undecalcified ground slides were used for Villanueva–Goldner staining [23] and examined with an optical microscope (Nikon ECLIPSE Ci, Nikon, Tokyo, Japan).

### 2.8. Statistical Analysis

The statistical software SAS 9.4 (SAS Institute Inc., Cary, NC, USA) was used for the evaluation of bone bonding in the two groups. The Mann–Whitney U test was used for the statistical analysis, and *p* < 0.05 (both sides) was considered significant.

## 3. Results

### 3.1. Evaluation of Bone Bonding

The MAH group had stronger bone-bonding than the untreated (UN) group for the entire duration of observation. The difference in bone bonding between the two groups was not significant at 1 and 8 weeks after implantation: 2.73 ± 1.49 N and 1.88 ± 1.21 N for the MAH group and UN group, respectively, at 1 week, and 23.86 ± 14.20 N and 19.23 ± 9.45 N for the MAH group and UN group, respectively, at 8 weeks. As shown in Figure 4, on the other hand, the MAH group showed significant increases in bone bonding at 4 and 12 weeks after implantation: 16.87 ± 9.90 N and 7.07 ± 2.91 N for the MAH group and UN group, respectively, at 4 weeks, and 49.31 ± 16.58 N and 17.32 ± 8.86 N for the MAH group and UN group, respectively, at 12 weeks.

### 3.2. Scanning Electron Microscopic Observation

As observed with scanning electron microscopy (SEM), Ca and P were present on the surface of the Ti samples separated from the calvarial bone. In addition, the UN group showed deposition of Ca and P mainly around the screws. However, bone formation also occurred at sites away from the screw hole in the MAH group, and Ca growth was observed in parallel with a longer implantation period. As shown in Figure 5 and Figure 6, moreover, a wider range of bone formation was observed on the side of the bone-contacting surface in the MAH group than in the UN group.

### 3.3. Histological Observation

As shown in Figure 7, the sagittal section of the extracted tissue at the joint was examined histologically. At 1 week after implantation, scattered resorption, likely due to osteoclasts, was observed on the bone surface in both the MAH and UN groups. At 4 weeks, active formation of new bone was observed in the MAH group, and activated round osteoblasts, were found on the bone surface. In addition, the surface of the AM Ti plates appeared rough, and a highly rough surface was observed in the MAH group. Furthermore, active bone formation between the calvarial bone and the Ti surface was seen in the MAH group. As shown in Figure 8, the gap between the bone and the Ti surface was completely filled with newly formed bone in the MAH group at 12 weeks, showing active bone formation from the Ti surface toward the deep part. On the other hand, in the UN group, bone formation on the Ti surface was slight even at 12 weeks. The MAH group showed osteoid bone formation at 4 weeks. On the other hand, in the UN groups, osteoid bone formation was not observed. No obvious inflammatory cell infiltration was observed in either group during the observation period.

## 4. Discussion

In this study, AM Ti plates showed strong bonding with the bone surface and osteogenic activity. In addition, the MAH treatment on the bone surface appeared to promote bone bonding and formation. Kawaii et al. [20,24] implanted a rolled Ti plate treated with MAH in the femur of a rabbit to evaluate bone bonding. In contrast, bone bonding of AM Ti plates treated with MAH under the condition of being placed on the bone surface was mechanically evaluated in the present study. In addition, the bone formation process between the AM Ti plate surface and the cortical bone was observed histologically. To the best of our knowledge, this is the first study to report such evaluations.

The development of fully customized artificial restorations to repair bone defects has been reported in recent years, expanding the scope of treatment in the fields of bone reconstruction, ossification, and rehabilitation [25,26]. Ti is also considered one of the biorepair materials due to its high bioaffinity. In addition, to produce customized artificial restorations by a metal modeling machine, their 3D data, which is generated by computer-aided design based on patient-specific CT and magnetic resonance imaging data using surgical simulation software, is converted into STL files. Additive manufacturing that creates complex shapes based on 3D data includes SLM, powder bed fusion that selectively melts and bonds a specific area of the powder bed by thermal energy represented by [27,28], and directed energy deposition that melts and bonds the powders by concentrating thermal energy while supplying powder material.

However, bone formation at the joint of the AM Ti device and its bonded bone surface has not been investigated [29]. The present study showed that the AM Ti plate has the same high bioaffinity in the living body after implantation as other Ti devices used clinically. Inoue et al. [16] used the patient-matched osteogenic Ti plate, which had been produced by SLM, for the bone defect at the dental implant placement site and reported good clinical progress in all cases. The present study aimed to demonstrate that the AM Ti plate bonds to the bone surface by ossification, and that osteogenic activity can be increased by MAH treatment of the Ti plate.

Clinically, conventional reconstructive plates do not have osteogenic activity and are mainly screw-fixed with mechanical bonding. However, the difficulty in matching the plates to the defect area and the fracture of the plate after reconstructive surgery are considered to be derived from the mechanical bonding. To solve these problems, we have been investigating the development of repair devices with a treated Ti surface that enables its integration with the bone. Thus, we previously evaluated the osteogenic activity in a study in which a rat calvarial bone defect was covered with the Ti mesh subjected to the MAH treatment, as well as other surface treatments. The results showed that the MAH-treated Ti mesh exhibited the highest bone formation in the bone defect area from an early stage, demonstrating the effectiveness of the MAH treatment on the Ti mesh [30,31,32].

In the present study, the AM Ti plate was firmly fixed to the surface of the rat calvarial bone with screws in accordance with the usage of the Ti plates in the actual bone reconstruction surgery. Subsequently, the mechanical strength of the bonding between the AM Ti and the bone surface 4, 8, and 12 weeks after implantation was compared between the MAH and UN groups. In addition, the bone formation process between the Ti and the bone surface was examined microstructurally and histologically. The results showed that the AM Ti subjected to the MAH treatment had strong bonding with cortical bone, which is the bone surface.

The reason why MAH treatment exhibits high osteogenic activity in Ti may be explained by the study conducted by Kokubo et al. [33,34,35]. They immersed the Ti metal plate treated with MAH in simulated body fluid to evaluate the deposition of apatite. When it was heated to 600 °C or higher after acid treatment, phosphate ions were preferentially absorbed on the Ti surface. Subsequently, calcium ions were adsorbed on the surface to deposit a layer of calcium phosphate, and crystalline apatite of calcium phosphate, which appeared as calcium phosphate, was further adsorbed, showing that the MAH treatment results in high osseous conduction and bone formation. Kawai [36] reported that the titanium hydride likely transformed into the rutile during the heat treatment according to the TF-XRD and porosity measurement. The MAH Ti surface became average peak to valley distance and the highest peak to the lowest valley distance was higher than the UN Ti Surface. In addition, it was reported that the rough Ti surface caused by MAH treatment was also involved in promoting bone formation in vivo. Moreover, with the bone formation toward the AM Ti plate, the distance between the bone surface and the AM Ti decreased over time in both groups. The bone bonding to the Ti surface was observed in both groups 8 and 12 weeks after implantation. In the evaluation of bone bonding, stronger bone-bonding was seen in the MAH group at 8 and 12 weeks after implantation. In the whole period of all observations, the MAH-treated Ti plate showed stronger bone bonding. Active formation of new bone was observed from the surface of the AM Ti toward the deep part, which was more evident in the MAH group. In addition, the histological evaluation showed no obvious inflammatory cell infiltration around the additively manufactured plate in either group. Thus, the bone bonding was stronger in the MAH group than in the UN group, likely due to the increase in chemical and physical bonding caused by the activation of bone formation. It is considered that the increase in bone bonding in the early postoperative period was caused by chemical bonding, and then the bonding was strengthened by the increase in physical fitting due to bone formation on the Ti surface. Therefore, the present findings suggest that MAH treatment is an effective surface treatment for AM Ti.

In the clinical setting, there is no bone-fixing plate that bonds to the bone, and it is common to fix plates mechanically with screws. However, this has problems such as plate fracture due to its incompatibility with the bone [37,38]. The causes of such problems likely include insufficient compatibility between the curved plate and the reconstruction site due to manipulation, micro-vibration caused by the loss of firm fixation due to invasion of soft tissues at the contact site between the bone and Ti device, and the added functional load of movement such as mastication in the mandible [39].

In this study, MAH treatment, which improves bone bonding, was applied to a fully customized, additively manufactured plate, which enables precise adaptation to bone defect morphology. Then, mechanical improvement of its bone bonding was shown, strong bone bonding to the treated surface was observed microstructurally, and a histological comparison to the UN group was performed, in which rapid and active bone formation, as well as new bone formation from the surface of the AM Ti toward the deep part, was observed. These results are considered to demonstrate the technologies that will lead to solutions for plate problems encountered in bone reconstruction surgery using Ti plates, attracting a great deal of attention in the fields of orthopedics, plastic surgery, neurosurgery, and oral surgery. Practical use of these technologies is expected in the future.

## 5. Conclusions

This study showed that the AM Ti artificial bone has the same high bioaffinity in the living body after implantation as other Ti devices, and that MAH treatment induces high osteogenic activity. Further research is expected to establish the effectiveness of the MAH-treated AM Ti plate for the reconstruction of bone.

## Figures and Tables

**Figure 1 materials-13-05104-f001:**
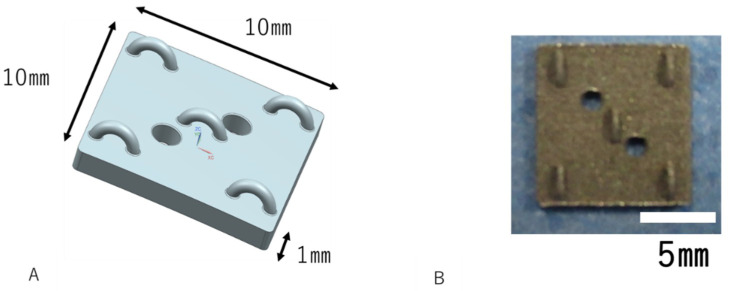
Shape of the Ti plates. (**A**): Designed plate. The dimensions of the AM Ti Plate samples are 10 mm × 10 mm × 1 mm. (**B**): Real image of an AM Ti Plate. The scale bar indicates 5 mm.

**Figure 2 materials-13-05104-f002:**
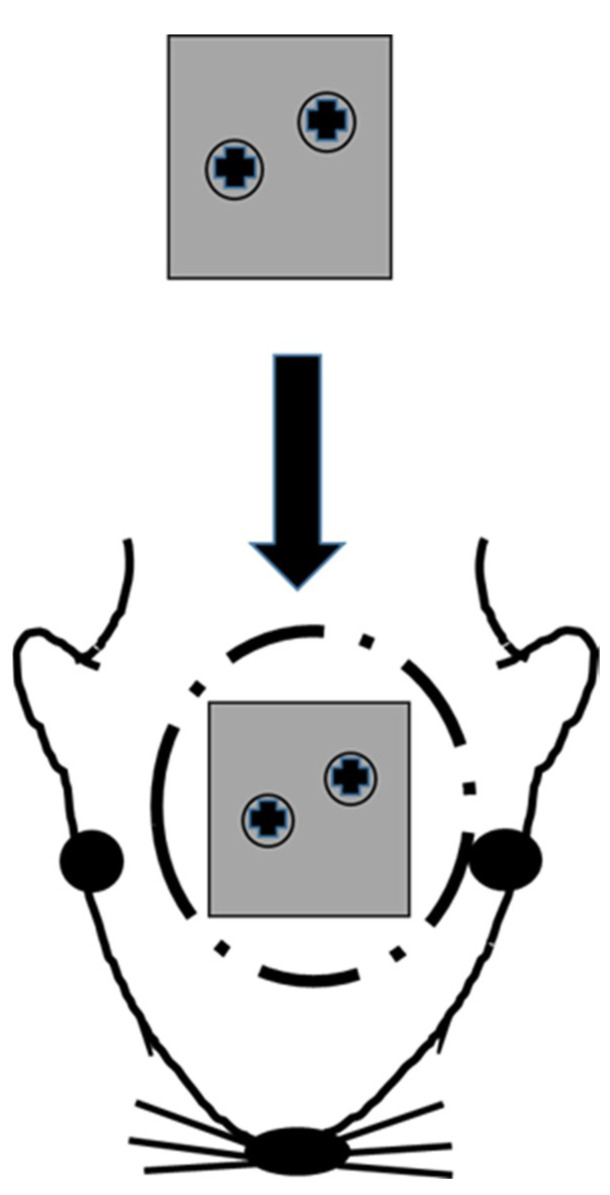
Implantation technique. The AM Ti plate is fixed to the calvarial bone.

**Figure 3 materials-13-05104-f003:**
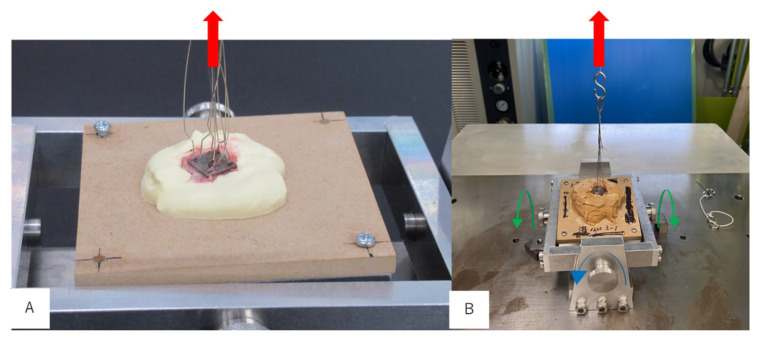
Evaluation of bone bonding. The bonding between the AM Ti plate and the cortical bone is evaluated by measuring the fracture load (N). The AM Ti plate is pulled out from the calvarial bone. (**A**): The lateral view. (**B**): The frontal view.

**Figure 4 materials-13-05104-f004:**
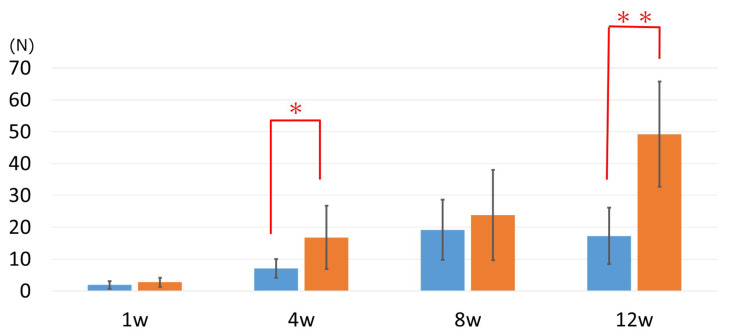
Evaluation of bone bonding. The mixed-acid and heat (MAH)-treated group shows significant increases in bone bonding at 4 and 12 weeks after implantation: 16.87 ± 9.90 N and 7.07 ± 2.91 N for the MAH group and the untreatedgroup, respectively, at 4 weeks, and 49.31 ± 16.58 N and 17.32 ± 8.86 N for the MAH group and the UN group, respectively, at 12 weeks. * <0.05, ** <0.01.

**Figure 5 materials-13-05104-f005:**
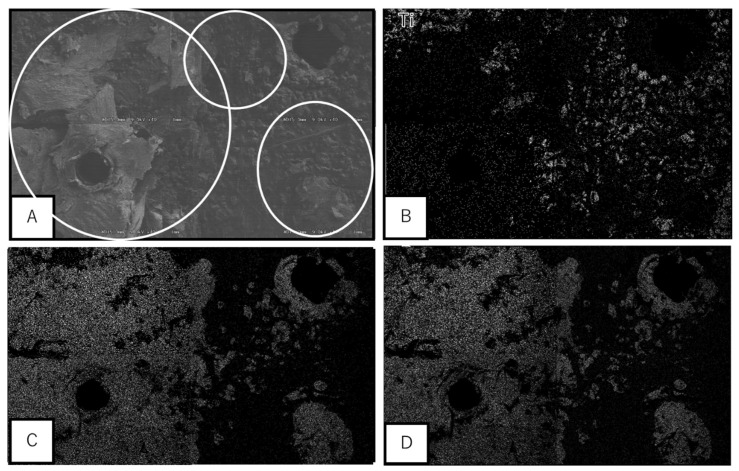
Electron micrograph by energy dispersive X-ray mapping. White circles indicate the area of bone tissue. (**A**): Backscattered electron image, (**B**): Titanium, (**C**): The area of calcium, (**D**): The area of phosphorus.

**Figure 6 materials-13-05104-f006:**
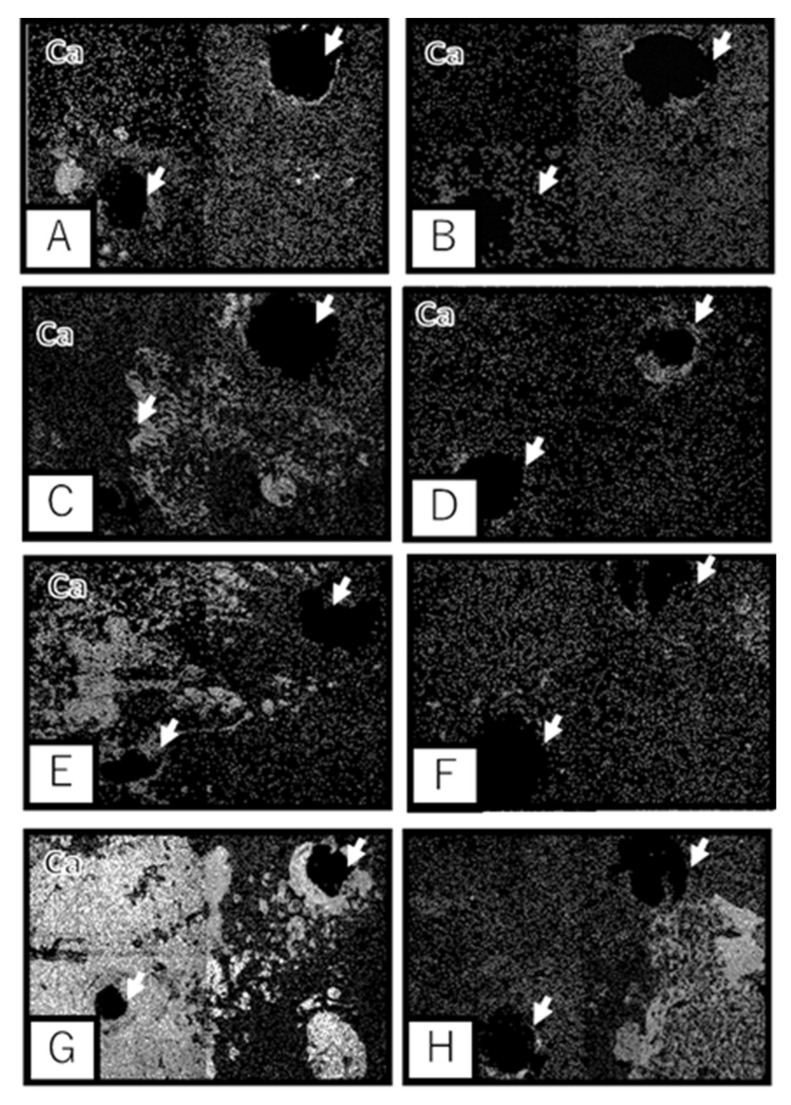
Scanning electron microscopic observation. Electron micrograph of Ca distribution by energy dispersive X-ray mapping of selective laser melting Ti metal. Arrows indicate screw holes. (**A**): The MAH group at 1 week. (**B**): The UN group at 1 week. (**C**): The MAH group at 4 weeks. (**D**): The UN group at 4 weeks. (**E**): The MAH group at 8 weeks. (**F**): The UN group at 8 weeks. (**G**): The MAH group at 12 weeks. (**H**): The UN group at 12 weeks.

**Figure 7 materials-13-05104-f007:**
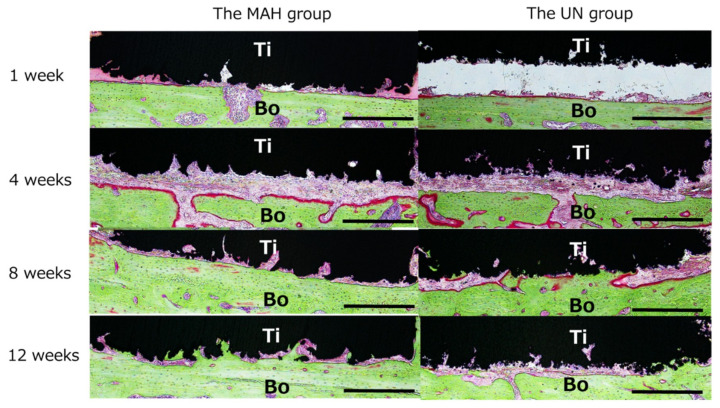
Histological observation. Active bone formation between the calvarial bone and the Ti surface is seen in the MAH group. The scale bar indicates 500 µm. Ti: Titanium plate, Bo: Calvarial bone.

**Figure 8 materials-13-05104-f008:**
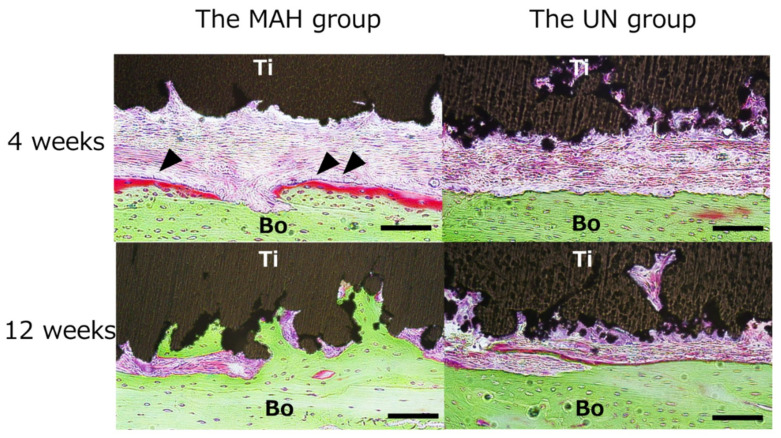
Enlarged histological observation. The MAH group at 12 weeks, showing active bone formation from the Ti surface toward the deep part. The scale bar indicates 100 µm. Ti: Titanium plate, Bo: Calvarial bone.

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
