# Peer review of "Mechanical, Histological, and Scanning Electron Microscopy Study of the Effect of Mixed-Acid and Heat Treatment on Additive-Manufactured Titanium Plates on Bonding to the Bone Surface"

_materials, 2020, doi:10.3390/ma13225104_

Round 1
Reviewer 1 Report
Interesting paper, but has to be revised before acceptance.
The clinical parts in introduction and discussion do not fit well in the manuscript and are far away from the question of the study.
Language has to be corrected by a native speaker.
2 authors for statistics of this not very complex study are not adequate.
Points of criticism in detail:
please check the precise name of the staining you used: Did you use the Masson-Goldner Trichome staining or van Gießon Staining?
Line 40: using autologous bone creates a donor side morbidity, consider adding the aspect
Line 51: Does Titanium also have disadvantages? If yes, what are strategies to overcome them?
Line 109: Specify the used device.
Line 132: Add the state time
Line 142: Check the Staining, as mentioned above
Line 155: Please specify tending
Fig. 7 and 8: The figures are in a quite low resolution, making it hard to differentiate the structures.
Fig 7 check the alignment of the pictures
Line 220: Resorbable and non-resorbable scaffold materials mixed up – what exactly is the future use of titanium?
Line 233: Specify the “good progress”.
Lines 274-276:
Please check your citations, e.g.:
Citation no 41. Probst, F.; Mast G.; Ermer M.; Gutwald R.; Scbmelzeisem R.; Pautke C.; Otto S.; Scbiel S.; Ebrenfeld M.;Cornelius C.; Metzger M. MatrixMANDIBLE Preformed reconstruction plates—a two-year two- institution experience in 71 patients. J Oral Maxillofac Surg 2012, 70, 657-666. DOI: 10.1016/j.joms.2012.06.175.
In your Paper, Line 275 you write: However, this has problems such as plate fracture due to its incompatibility with the bone [40,41]
Looking into the Paper of Probst et al.,
“Of the 70 subjects, no plate failure because of fatigue fractures was noted.” (one patient didn´t get a MMPRP (MatrixMANDIBLE Preformed Reconstruction Plate) because of anatomic variations and was excluded from the analysis.
Yes, Probst et al. mentions in his introduction, that fatigue fractures are one of the most common complications of reconstruction plates. But at the same time, he shows, that none of is treated patients suffered from a plate fracture. If you would like to outline the problem of plate fractures, I would suggest citing a paper reporting on the incidence of plate fractures.
Citation no 8. Smeets, here first and family names of the authors are confused
Author Response
We wish to express our appreciation to the reviewers for their insightful comments on our paper. The comments have helped us significantly improve the paper
Point1:The clinical parts in introduction and discussion do not fit well in the manuscript and are far away from the question of the study.
Responce1:We thought that MAH treatment was effective surface modification for the titanium. The early formation and strong bone bonding is required in clinical situation.
We added the sentence in introduction.
“We considered that The AM Ti plate with MAH treatment perform strong bone bonding and early bone formation, MAH treatment became effective approach for the plate fracture and infection.”
Point2: Language has to be corrected by a native speaker.
Responce2:The paper has been edited and rewritten by an experienced scientific editor, who has improved the grammar and stylistic expression of the paper.
Point3:2 authors for statistics of this not very complex study are not adequate.
Responce3:We corrected the section of statics analysis for one authors.
Points of criticism in detail:
Point4:please check the precise name of the staining you used: Did you use the Masson-Goldner Trichome staining or van Gießon Staining?
Responce4: We prepared non-decalcified polished specimens, stained them with Villanueva-Goldner stain, and observed them with an optical microscope. We thought the Figure 7 and 8 was too bright and retaken the figure which is adjustmented to light . We added the reference in this section which explain the Villanueva-Goldner stain .
Point5: Line 40: using autologous bone creates a donor side morbidity, consider adding the aspect
Responce5: We reconsider the sentence.
“this requires highly advanced surgical techniques, as well as extended surgery, and the amount of autologous bone used is limited.”
to
“this requires highly advanced surgical techniques, as well as the amount of autologous bone used is limited, and the donor site surgical infection is concerned.”
Point6:Line 51: Does Titanium also have disadvantages? If yes, what are strategies to overcome them?
Response:6:Titanium have high biocompativility. However, Follow a significant clinical course once an plate infection ,fracture and Loosening has occurred. We considered that strong bone bonding and early bone formation is required in the Ti Plate.
Point7: Line 109: Specify the used device.
Responce7:We have changed the sentence referring the device.  
Point8: Line 132: Add the state time
Responce8: Accordingly, we have added the state time to this section.
Point9: Line 142: Check the Staining, as mentioned above
Responce9: please check the above answer.
Ponit10: Line 155: Please specify tending
Responce10: ” tend” was an incorrected word. There was significance difference between MAH and UN groups. 
Point11 ,12: Fig. 7 and 8: The figures are in a quite low resolution, making it hard to differentiate the structures.
Fig 7 check the alignment of the pictures
Responce11,12 :We thought the Figure 7 and 8 was too bright and retaken the figure which is adjustmented to light .
Point13: Line 220: Resorbable and non-resorbable scaffold materials mixed up – what exactly is the future use of titanium?
Responce13:This sentence meant “titanium is one of the biorepair materials”
The sentence of explaining tricalcium phosphate (β-TCP) and hydroxyapatite (HA) was deleted.
Point14: Line 233: Specify the “good progress”.
Responce14: In accordance with the reviewer's comment, we have changed “good progress”to “good clinical progress”
Point15:Lines 274-276:
Please check your citations, e.g.:
Citation no 41. Probst, F.; Mast G.; Ermer M.; Gutwald R.; Scbmelzeisem R.; Pautke C.; Otto S.; Scbiel S.; Ebrenfeld M.;Cornelius C.; Metzger M. MatrixMANDIBLE Preformed reconstruction plates—a two-year two- institution experience in 71 patients. J Oral Maxillofac Surg 2012, 70, 657-666. DOI: 10.1016/j.joms.2012.06.175.
In your Paper, Line 275 you write: However, this has problems such as plate fracture due to its incompatibility with the bone [40,41]
Looking into the Paper of Probst et al.,
“Of the 70 subjects, no plate failure because of fatigue fractures was noted.” (one patient didn´t get a MMPRP (MatrixMANDIBLE Preformed Reconstruction Plate) because of anatomic variations and was excluded from the analysis.
Yes, Probst et al. mentions in his introduction, that fatigue fractures are one of the most common complications of reconstruction plates. But at the same time, he shows, that none of is treated patients suffered from a plate fracture. If you would like to outline the problem of plate fractures, I would suggest citing a paper reporting on the incidence of plate fractures.
Responce15: We have changed references which reporting on the incidence of plate fractures.
Almansoori, AA.; Choung, HW.; Kim, B.; Park, JY.; Kim, SM.; Lee, JH. Fracture of Standard Titanium Mandibular Reconstruction Plates and Preliminary Study of Three-Dimensional Printed Reconstruction Plates. J Oral Maxillofac Surg 2020, 78, 153-166. DOI: 10.1016/j.joms.2019.07.016.
Point16: Citation no 8. Smeets, here first and family names of the authors are confused
Responce16: This error has been correcte

Reviewer 2 Report
It is a very well written paper, well designed and its main subject is interesting and rather prototype.
Author Response
Thank you for confirming the manuscript.
We wish to express our appreciation to the reviewer for your insightful comments, which have helped us significantly improve the paper.
Kind regards
Reviewer 3 Report
The article describes bonding and osseointegration of rat calvarial bone to the surface of titanium plate manufactured by selective laser melting and additionally treated by acid etching and heating.
About the article I have some questions and comments:
Titanium implants and their relationship to biocompatibility and osseointegration is highly dependent on roughness and wettability of surfaces in contact with bone. You should to add some evaluation of these parameters and compare MAH and UN.
Line 91: The ratio between sulfuric acid and hydrochloric acid should be stated.
Line 142: Are you sure with V. Goldnar staining? Is it not Goldner’s trichrome staining?
On Figure 4 should be the significance marked with an asterisk (* <0.05, ** <0.01, *** <0.001).
What do you think why the bone bonding is not significant at 8 weeks after implantation similar to weeks 4 and 12? And why at week 8 is the bone bonding of UN samples higher than after next four weeks? Is this difference significant?
In legend of figure 7 or 8 should be described the type of tissue (mineralized, osteoid, etc.)
Line 292: Better will be … osteogenic activity than capacity.
Line 302: In the Conflict of Interest section, it should be noted that one of the authors is employed by a private company producing the evaluated material.
Author Response
We wish to express our appreciation to the reviewers for their insightful comments on our paper. The comments have helped us significantly improve the paper
Point1:Titanium implants and their relationship to biocompatibility and osseointegration is highly dependent on roughness and wettability of surfaces in contact with bone. You should to add some evaluation of these parameters and compare MAH and UN
Responce1:We agree that additional information on roughness and wettability as the reviewer suggested would be valuable.
We added this sentence and reference.
“Kawai[23] reported that The titanium hydride likely transformed into the rutile during the heat treatment according to the TF-XRD and porosity measurement. The MAH Ti surface became average peak to valley distance and the highest peak to the lowest valley distance is higher than the UN Ti Surface. “
Point2:Line 91: The ratio between sulfuric acid and hydrochloric acid should be stated.
Responce2:I added in this sentence. “the ratio of the mixed-acid solution is 5 ml of the mixed-acid solution of a 1:1(w/w) mixture of 66.3% H2SO4 (w/w) solution and 10.6% HCl (w/w) solution. “
Point3: Line 142: Are you sure with V. Goldnar staining? Is it not Goldner’s trichrome staining?
Responce3: We prepared non-decalcified polished specimens, stained them with Villanueva-Goldner stain, and observed them with an optical microscope. We thought the Figure 7 and 8 was too bright and retaken the figure which is adjustmented to light. We added the reference in this section which explain the Villanueva-Goldner stain .
Point4:On Figure 4 should be the significance marked with an asterisk (* <0.05, ** <0.01, *** <0.001).
Response: We have added an asterisk on Figure4.
Point5: What do you think why the bone bonding is not significant at 8 weeks after implantation similar to weeks 4 and 12? And why at week 8 is the bone bonding of UN samples higher than after next four weeks? Is this difference significant?
Responce5: The difference of the bone bonding between UN weeks 8 and 12 was not significant. The reason for why weeks 8 UN groups bone-bonding is higher than 12 weeks was rats individual difference. We considered that UN sample bone was slightly ingrowth for the titanium surface after 8weeks ,on the other hand MAH was significant ingrowth.
Point6: In legend of figure 7 or 8 should be described the type of tissue (mineralized, osteoid, etc.)
Responce6:we added the sentence which explain the osteoid bone formation in the 4 weeks MAH and UN groups.
Point7:Line 292: Better will be … osteogenic activity than capacity.
Responce7:In accordance with the reviewer's comment, We have converted the word “osteogenic” capacity to the word “osteogenic activity”
Point8:Line 302: In the Conflict of Interest section, it should be noted that one of the authors is employed by a private company producing the evaluated material.
Responce8:We ordered the production of Ti sample from Osaka Yakin Kogyo Co., Ltd. Ti sample was designed and produced in Osaka Yakin Kogyo Co., Ltd. We declared that no conflict of interests exists between Osaka medical college and Osaka Yakin Kogyo Co., Ltd.
